# The mouse suprachiasmatic nucleus encodes irradiance via a diverse population of neurons monotonically tuned to different ranges of intensity

Patrycja Orlowska-Feuer[1] , Beatriz Bano-Otalora[2] , Jessica Rodgers[1] , Franck P. Martial[1], Riccardo Storchi[1] and Robert James Lucas[1]

[1]*Division of Neuroscience, School of Biological Sciences, Faculty of Biology, Medicine and Health, University of Manchester, Oxford Road, Manchester, UK*

[2]*Centre for Biological Timing, Faculty of Biology, Medicine and Health, University of Manchester, Oxford Road, Manchester, UK*

Handling Editors: Katalin Toth & Nathan Schoppa

The peer review history is available in the Supporting Information section of this article (https://doi.org/10.1113/JP285000#support-information-section).

**Abstract**   Many neurons of the mammalian master circadian oscillator in the suprachiasmatic nuclei (SCN) respond to light pulses with irradiance-dependent changes in firing. Here, we set

**Patrycja Orlowska-Feuer** received her PhD in Biology from the Jagiellonian University in Krakow, Poland, in 2014. Following a postdoctoral position at the Jagiellonian University, she obtained Bekker Fellowship (NAWA, Poland) and subsequently a Marie Sklodowska-Curie Individual Fellowship and became a research fellow at the University of Manchester. Her current research focuses on comparing light-induced neuronal activity within the suprachiasmatic nucleus and more general hypothalamic area between diurnal and nocturnal rodents.

The Journal of Physiology

out to better understand this irradiance coding ability by considering how the SCN tracks more continuous changes in irradiance at both population and single unit level. To this end, we recorded extracellular activity in the SCN of anaesthetised mice presented with up + down irradiance staircase stimuli covering moonlight to daylight conditions and incorporating epochs with steady light or superimposed higher frequency modulations (temporal white noise (WN) and frequency/contrast chirps). Single unit activity was extracted by spike sorting. The population response of SCN units to this stimulus was a progressive increase in firing rate at higher irradiances. This relationship was symmetrical for up *vs.* down phases of the ramp in the presence of white noise or chirps but exhibited hysteresis for steady light, with firing systematically higher during increasing irradiance. Single units also showed a monotonic relationship between firing and irradiance but exhibited diversity not only in response polarity (increases *vs.* decreases in firing), but also in the sensitivity ($EC_{50}$) and slope of fitted functions. These data show that individual SCN neurons exhibit monotonic relationships between irradiance and firing rate but differ in the irradiance range over which they respond. This property may help the SCN to encode the large differences in irradiance found in nature using neurons with a constrained range of firing rates.

(Received 18 May 2023; accepted after revision 11 September 2023; first published online 30 September 2023)

**Corresponding authors** P. Orlowska-Feuer and R. J. Lucas: Division of Neuroscience, School of Biological Sciences, Faculty of Biology, Medicine and Health, University of Manchester, Manchester, UK. Email: patrycjaanna.orlowska-feuer@manchester.ac.uk and robert.lucas@manchester.ac.uk

**Abstract figure legend** This study introduces staircase light stimuli mimicking changes in ambient light intensities across night to day transition with the inclusion of more naturalistic, high temporal frequency modulations presented at each step. It has been used here to probe irradiance coding properties of the suprachiasmatic nucleus (SCN) at population and single cell level in anaesthetised mouse. We observed monotonic relationships between firing rate and irradiance at both population and single cell level, although individual neurons track different portions of irradiance range. Thus, we provide evidence of the existence of sparse irradiance code at the level of the SCN. Figure created with BioRender.com.

### Key points

- Daily changes in environmental light (irradiance) entrain the suprachiasmatic nucleus (SCN) circadian clock. The mouse SCN shows graded increases in neurophysiological activity with light pulses of increasing irradiance.
- We show that this monotonic relationship between firing rate and irradiance is retained at population and single unit level when probed with more naturalistic staircase increases and decreases in irradiance.
- The irradiance response is more reliable in the presence of ongoing higher temporal frequency modulations in light intensity than under steady light.
- Single units varied in sensitivity allowing the population to cover a wide range of irradiances.
- Irradiance coding in the SCN has characteristics of a sparse code with individual neurons tracking different portions of the natural irradiance range. This property may address the challenge of encoding a $10^9$-fold day:night difference in irradiance within the constrained range of firing rates available to individual neurons.

## Introduction

The suprachiasmatic nucleus (SCN) of the hypothalamus plays a central role in co-ordinating 24 h rhythms in metabolism, physiology and behaviour by detecting day/night changes in environmental light intensity and translating that into changes in the phase, waveform and amplitude of circadian rhythms. Light information is conveyed to the mammalian SCN by a direct retinohypothalamic projection (Abrahamson, 2001) comprised primarily of the M1 type of melanopsin expressing intrinsically photosensitive retinal ganglion cells (ipRGCs) (Baver et al., 2008; Li & Schmidt, 2018; Stinchcombe et al., 2021).

SCN neurons respond to ocular light exposure with either increases or decreases in spike firing (Groos &

Mason, 1978; Nishino et al., 1976), indicating that the time of day signal (ambient light) may be encoded in the firing rate of SCN neurons. This possibility in turn raises the question of how the $\sim10^9$-fold change in irradiance across a day is represented in firing rates which may vary by only two to three orders of magnitude. Many studies have applied light pulses of differing intensity to rodents and described a monotonic relationship between stimulus irradiance and SCN firing rate at population (Brown et al., 2011; Dobb et al., 2017; Groos, 1980; Meijer et al., 1998; van Diepen et al., 2013; van Oosterhout et al., 2012; Walmsley & Brown, 2015) and, in some cases, single unit (Brown et al., 2011; Dobb et al., 2017; Walmsley & Brown, 2015) levels. A parsimonious interpretation of those findings is that light responsive SCN neurons represent the diurnal irradiance change with progressive alterations in time average firing. However, there is recent evidence indicating that irradiance coding might be quite different in the retinal input to the SCN and/or when probed with more naturalistic stimuli. Thus, when presented with staircase changes in irradiance, only a minority of the M1 ipRGCs that dominate input to the SCN show the predicted monotonic increase in firing (Liu et al., 2023; Milner & Do, 2017). Most show so-called unimodal tuning, in which firing rate peaks at an intermediate irradiance, falling with further increases in light intensity (Liu et al., 2023; Milner & Do, 2017).

The literature thus provides support for two distinct modes of irradiance coding in the retina-SCN projection.

SCN recordings raise the possibility of what may be called a 'distributed code' in which a uniform population of neurons each expresses gradual changes in firing across the full natural irradiance range. Conversely, recordings of M1 RGCs suggest a 'sparse' code in which individual neurons encode restricted irradiance ranges. It is tempting then to infer that the 'sparse' irradiance code of retinal input is transformed to a 'distributed' code in the SCN. However, an important caveat is the lack of data regarding the SCN response to the sort of continuous changes in irradiance used by Liu et al. (2023) and Milner & Do (2017) to reveal unimodal irradiance relationships in retinal output. Indeed, M1 ipRGCs show monotonic irradiance responses when presented with light pulses (Kofuji et al., 2016; Zhao et al., 2014) raising the possibility that the apparent differences in irradiance coding between ipRGCs and SCN are rather a consequence of the type of visual stimulus employed.

We set out here therefore to record mouse SCN responses to continuous changes in irradiance of the type represented by the staircase stimulus of Milner & Do (2017). In an attempt to make this stimulus more naturalistic, we modified it to include ongoing visual stimuli. Light intensity within individual ipRGC receptive fields is expected to be constantly changing during natural viewing as a result of not only ongoing alterations in ambient light, but also changes in direction of view across scenes rich in spatial contrast (Dobb et al., 2017). We have previously shown that such higher frequency modulations in light intensity are represented in the firing pattern of the

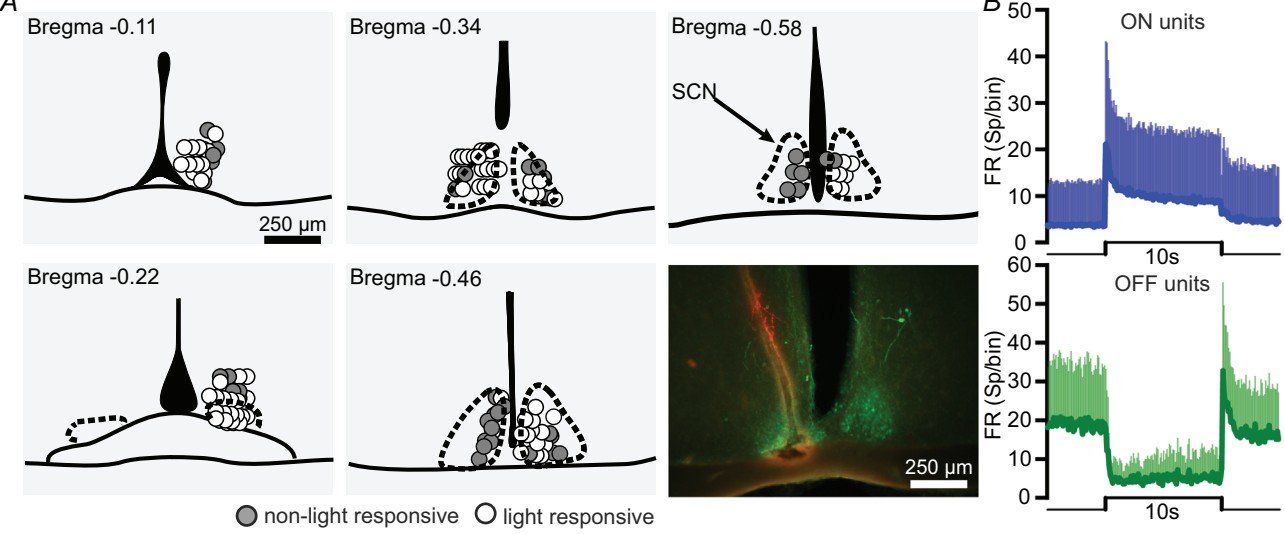

**Figure 1. Light responsive neurons in the mouse SCN**
*A*, schematic sequential coronal slices through the SCN with estimated anatomical localisation of recorded neurons, colour coded depending on whether they were light responsive (white) or not (grey), followed by an exemplar image showing electrode trace (CM-Dil fluorescent dye in red) in the SCN (AVP staining in green) for one of the recordings. *B*, mean PSTHs + SD (bin: 0.1 s) for all units excited (50/57) or inhibited (7/57) by light (top and bottom trace, respectively). [Colour figure can be viewed at wileyonlinelibrary.com]

mouse SCN, and that they can have minor effects on time averaged firing (Dobb et al., 2017; Mouland et al., 2017). We therefore included two types of ongoing modulation at each step in the irradiance staircase [a high contrast temporal white noise (WN) and temporal frequency and contrast chirps], designed to probe SCN activity across a range of contrasts and rates of change.

## Methods

### Animals

All experiments were performed in accordance with the Animals, Scientific Procedures Act of 1986 (United Kingdom) approved by Home Office (PP3176367). In total, five C57BL/6 adult mice were used (University of Manchester; three females and two males, 4−6 months old). Animals were group-housed under strict 12:12 h light/dark photocycle at 22°C with water and food available *ad libitum*.

### *In vivo* electrophysiology

**Surgery.** In preparation for stereotaxic surgery, mice were removed from their home cages 2−3 h prior to lights off and were subsequently anaesthetised by I.P. injection of urethane (1.4–1.5 g kg$^{-1}$, 20% solution in sterile saline; Sigma-Aldrich, St Louis, MO, USA). The anaesthesia level was verified by the lack of withdrawal and ocular reflexes and an additional dose of urethane (10% of initial dose) was supplied if required. Atropine (0.3 mg kg$^{-1}$; Sigma-Aldrich) and saline were administrated S.C. to reduce mucous secretion and ensure adequate hydration, respectively. Throughout the experiment, the body temperature of animals was maintained at 37.0 ± 0.5°C using a homeothermic blanket system. Each animal's head was secured in a stereotaxic frame via ear and incisor bars. The skull was exposed and two stereotaxic points: bregma and lambda were set. The co-ordinates for the SCN were assessed based on stereotaxic brain atlas for mice (Paxinos & Franklin, 2004) and craniotomy was performed at 1.0 mm lateral and 0.1 mm caudal to bregma. A Buszaki 32L probe (Neuronexus, Ann Arbor, MI, USA) consisting of four shanks spaced 200 $\mu$m, each with eight recording sites, was used as a recording probe. It was dipped in a fluorescent dye (CM-DiI; V22888; Thermo Fisher Scientific, Waltham, MA, USA) before the first insertion for subsequent histological verification of recording places. The recording probe was positioned sagitally on an angle of 9° relative to the dorsal–ventral axis with the rostral shank at 0.8 mm lateral and 0.0 mm caudal to bregma. The electrode was slowly lowered down to the level of the SCN (4.5–5.5 mm from the brain surface) using a fluid-filled micromanipulator (MO-10; Narishige International Ltd, London, UK).

**Recordings.** The SCN was electrophysiologically recognised by the presence of light responsive neurons (the light test was performed when moving towards the SCN). Once light responsive units were found, animals were dark adapted for 30 min, which also allowed neuronal activity to stabilise. A Recorder64 system (Plexon Inc., Dallas, TX, USA) was used to acquired neuronal activity, which was amplified (×3000), high pass filtered at 300 Hz and digitised at 40 kHz. Once the recording was complete, the probe was moved either deeper or slowly raised and move lateral to sample more units from non-overlapping areas.

**Histology.** At the end of experiment, mice were intracardially perfused with saline followed by 4% paraformaldehyde. Brains were removed, postfixed overnight at 4°C and cryoprotected using 30% sucrose in phopshate-buffered saline (PBS). Brains were cut (40 $\mu$m) on a freezing microtome at a coronal plane. Immuno-fluorescence labelling for arginine vasopressin (AVP) was performed to delineate SCN anatomical boundaries. Briefly, following washes in 0.1 M PBS and 0.1% Triton X-100 in PBS, sections were blocked for 1 h in 5% normal donkey serum. Then, they were incubated with primary antibody (dilution 1:5000; AVP Rabbit; AB1565; Millipore, Burlington, MA, USA) for 2 days at 4°C. After washing, sections were incubated with Alexa Fluor 488 Donkey anti-Rabbit IgG (dilution 1:800; A-21 206; Invitrogen, Thermo Fisher Scientific) overnight at 4°C. Finally, sections were mounted onto gelatine coated slides and cover-slipped using ProLong Gold Antifade Mountant (Molecular Probes, Invitrogen; Thermo Fisher Scientific). Photos were taken using a DFC365 FX camera (Leica, Wetzlar, Germany) connected to a DM2500 micro-scope (Leica) using LAS AF6000 software (Leica). Recording sites were estimated based on stereotaxic notes, CM-DiI marks and AVP staining in accordance with atlas of Paxinos & Franklin (2001).

**Light stimuli.** A pE-4000 system (CoolLED, Andover, UK) connected to a liquid light guide fitted with a diffuser (Edmund Optics, Barrington, NJ, USA) was used as a light source. The diffuser was positioned at ∼5 mm from eye contralateral to the recording site. White light was used for all stimuli, consisting of output from four LEDs with peak emission at 385, 460, 550 and 635 nm. Irradiance was controlled by the motorised filter wheel (FW102C/FW212C Series; Thorlabs Inc., Newton, NJ, USA). A SpectroCAL MKII spectroradiometer (Cambridge Research System, Cambridge, UK) was used for light measurements. All irradiances are presented as

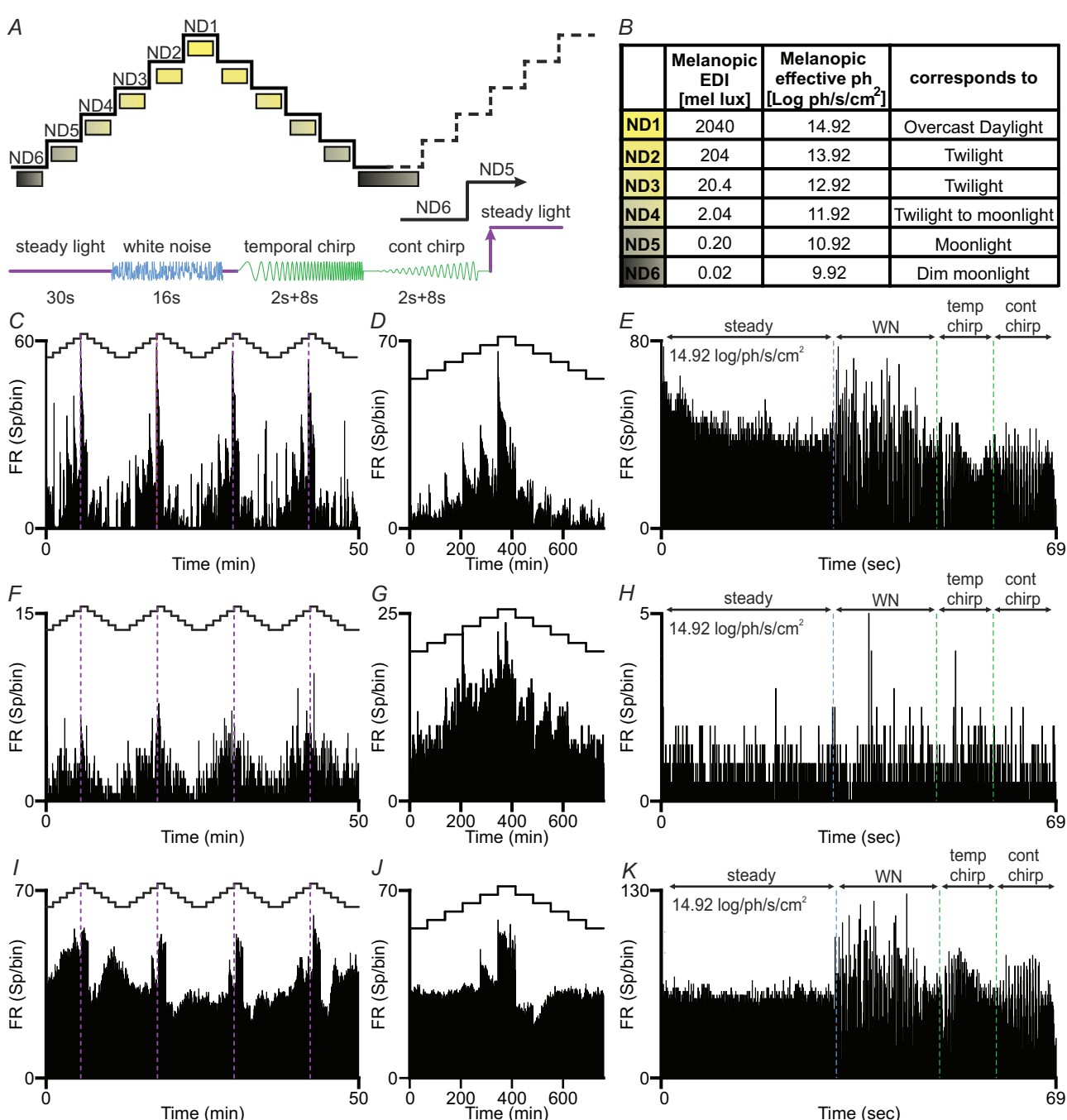

**Figure 2. Representative responses to the staircase stimulus**
*A*, schematic of a single presentation of the staircase stimulus showing (below) the composition of each step, consisting of steady light, WN, temporal + contrast chirps epochs and (above) the order of presentation for steps to an ascending and descending pyramid. Irradiance at each step indicating above as effective number of neutral density filters (ND); dotted line to right indicates start of next presentation. *B*, irradiance at each step of the staircase in melanopic EDI, melanopsin effective photon flux and descriptor for equivalent natural light condition. *C*, *F* and *I*, firing rate histograms (bin: 3s) of representative single SCN neurons across the multiple repeats of the staircase stimulus (schematic of stimulus shown above). *D*, *G* and *J*, PSTHs (bin: 1 s) for the staircase irradiance stimuli for cells presented in (*C*), (*F*) and (*I*), respectively. *E*, *H* and *K*, corresponding PSTHs (bin: 0.05 s) for steady, WN, temporal and contrast chirps at the highest irradiance step (14.92 log photons s$^{-1}$ cm$^2$ ) for cells presented in (*C*), (*F*) and (*I*), respectively. [Colour figure can be viewed at wileyonlinelibrary.com]

melanopsin effective photons (log photons s$^{-1}$ cm$^{-2}$) and melanopic equivalent daylight illuminance (EDI) (mel lux). To classify cells as light responsive 10 repeats of 10 s long bright full-field light step (15.3 log photons s$^{-1}$ cm$^{-2}$) was presented with an interval of 50 s. Staircase stimuli consisted of six ascending and following descending steps, and such a pyramid was repeated eight times. The light intensities used were: 9.92, 10.92, 11.92, 12.92, 13.92 and 14.92 log photons s$^{-1}$ cm$^{-2}$ for subsequent steps. Each step started with 30 s of steady light, followed by superimposed temporal WN for 16 s and a temporal chirp for 8 s, and finished with a contrast chirp for 8 s. In-between temporal WN and chirp and both chirps, 2 s of steady light was introduced. The mean light irradiance was kept constant at each step throughout. Temporal WN was presented at a rate of 20 Hz as a pseudorandom change in irradiance spanning between dark and twice the irradiance for the steady light intensity. The temporal element of chirp accelerated at rate of 1 Hz s$^{-1}$ in the range 1−8 Hz as a sinusoidal modulation between dark and 98% contrast. Contrast element of the chirp was presented as a 2 Hz sinusoidal modulation increasing from 3% to 98% contrast.

## Data analysis

**Spike sorting.** Spike sorting was first conducted by an automated template-matching based algorithm (Kilosort2) (Pachitariu et al., 2016). Next, identified clusters and multiunit spikes were exported to Offline Sorter (Plexon Inc., Dallas, TX, USA), as 'virtual tetrodes' (spike waveforms detected across four adjacent channels) for manual verification. Multivariate analysis of variance *F* statistics, J3 and Davies-Bouldin validity metrics, and interspike interval distribution (distinct refractory period >1.0 ms) were used to confirm single unit isolation.

**Light responsive unit classification.** Peristimulus time histograms and raster plots (PSTHs) (bin size = 0.1 s) were calculated in Neuroexplorer (Nex Technologies, USA). Light responsive units were identified based on the 2 SD rule, according to which responses were considered significant if, at the light onset (0–300 ms), offset (0–300 ms) or during the last 5 s of the stimulus, a cell's activity was higher or lower than the pre-stimulus firing rate (5 s) by 2 SD. A further manual screen was applied to exclude units passing this criterion as a result of changes in baseline activity rather than repeated responses to the stimulus. Based on the 2 SD rule, units were classified as ON units or OFF units.

**Irradiance coding.** The time averaged firing rate at each step and separately for steady (10 s), WN (16s) and chirp (20 s) epochs was computed for each unit and normalised to the unit's maximum time averaged firing rate observed across the staircase (giving 39 points for normalisation). This normalisation was applied across all values when normalised FR is shown. Activity of all units was carefully monitored by manual screening, and only stable single-unit activity was taken for further analysis (units had to keep stable firing rate throughout at least four presentations of the staircase protocol). Time averaged normalised firing rates were baseline subtracted and irradiance response curves were constructed by fitting four-parameter (Top, Bottom, logEC$_{50}$ and Slope) sigmoid curves with the minimum constrained to zero as appropriate. Units with a high goodness of fit ($r^2$) for this model for both ascending and descending steps (threshold was set at 0.85 based on the $r^2$ relationship with variability in FR across the staircase) were identified as neurons for which the behaviour could be approximated by a mono-tonic relationship with irradiance.

Equation for the sigmoidal curve

$$Y = \text{Bottom} + \frac{\text{Top} - \text{Bottom}}{1 + 10^{(\text{Log10EC}_{50} - \text{Log10Photons}) \times \text{Hill Slope}}}$$

where LogPhotons is the intensity in log photons s$^{-1}$ cm$^{-2}$.

Comparisons of sensitivity under various conditions were assessed by the *F* test for differences in EC$_{50}$, Hill slope and/or maxima.

**Statistical analysis.** Experimental data are presented as the mean ± SD. A Shapiro–Wilk test was performed to test for normal distribution. Group data was compared by a Friedman test or *F* test. Normalisation, irradiance sensitivity curve fitting, *F* test and Friedman test were performed in Prism, version 9.1.2 (Graphpad Software Inc., San Diego, CA, USA). The correlation coefficient between EC$_{50}$ and Hill slope for up and down phases was tested by using either Pearson (*r*) or Spearman (*r*$_s$) two-tailed correlation tests in Prism, version 9.1.2 (Graphpad Software Inc., San Diego, CA, USA).

## Results

We targeted the SCN of five anaesthetised mice with silicon multichannel recording electrode probes during their subjective night (i.e. when the SCN is most sensitive to light) (Brown et al., 2011; Meijer et al., 1998) according to stereotaxic coordinates. Of 84 single units estimated to fall within the boundaries of the SCN in these recordings, 57 showed a significant change in spike firing in response to a 10 s light step (Fig. 1*A*). The majority of these light responsive units were excited by light appearance (and in some cases disappearance) (*n* = 50/57) (Fig. 1*B*, top), with the remainder being inhibited during the 10 s light presentation (*n* = 7/57) (Fig. 1*B*, bottom). This diversity in light response polarity is consistent with

previous reports for SCN neurons (Brown et al., 2011; Groos & Mason, 1978; Meijer et al., 1998; Nishino et al., 1976; Orlowska-Feuer et al., 2020).

The main aim of the present study was to describe responses of the mouse SCN to continuous changes in irradiance at single cell resolution. We thus applied ascending and descending irradiance steps covering the range from dim moonlight to an overcast day (melanopic EDI between 0.02 and 2040 mel lux), repeated eight times (Fig. 2A and B). At each step, the animal was presented with 30 s of steady light, followed by 16 s of WN (20 Hz pseudorandom modulation between dark and twice the steady light intensity), 8 s of temporal chirp (sinusoidal modulation between 3% and 98% contrast at $1-8$ Hz, accelerating at 1 Hz s$^{-1}$) and 8 s of contrast chirp (sinusoidal modulation at 2 Hz increasing from 3% to 98% contrast) (Fig. 2A). The mean irradiance for each of these stimuli was kept the same as the irradiance of the steady light presented at the beginning of the step. Traces of three representative units, each of which appeared to respond to elements of the staircase stimulus (irradiance steps and/or superimposed higher frequency modulations), are presented in Fig. 2C–K.

As a first description of the response of the SCN to changing background light, we pooled the activity of individual light responsive units to describe the population-level response to this stimulus. The time averaged firing rate at each step (and separately for steady, WN and chirp epochs) was computed for each unit and normalised to the highest mean time averaged firing observed for that unit ($= 1$) across repeats of the staircase. Population means for these data revealed mono-

tonic relationships between irradiance and firing rate under each of the three stimulus conditions (Fig. 3A–C). Interestingly, the form of this relationship was different for ascending *vs.* descending phases of the staircase for the steady light (*F* test, $P = 0.0256$, $n = 57$) (Fig. 3A), but not when WN or chirp stimuli were superimposed (*F* test, chirp: $P = 0.3097$, $n = 57$; WN: $P = 0.3804$, $n = 57$) (Fig. 3B and C). In this way, irradiance coding was more reliable in the presence of ongoing visual stimuli. The mean $\Delta$FR between top and bottom of the staircase stimuli was not statistically different between conditions (Friedman test, $P = 0.1421$, $F = 3.903$, $n = 57$) (Fig. 3D).

At the population level, the mouse SCN thus encoded irradiance with a monotonic increase in time averaged firing rate under all conditions tested. Next, we looked at the pattern of activity for individual units, initially with particular attention to the possibility that SCN units recapitulated the unimodal irradiance response relationship reported for M1 ipRGCs (Liu et al., 2023; Milner & Do, 2017). We first filtered all SCN units ($n = 84$) to identify those that responded to elements of our staircase stimulus with changes in maintained firing rate. To allow for the possibility that such responses could be restricted to intermediate irradiances (unimodal) and/or to a subset of stimulus condition (e.g. WN or chirp), we classified any unit for which time-averaged FR at any of the five steps under any condition averaged across all repeats lay outside $\pm 2$ SD of that at the dimmest intensity (ND6, 9.92 log photons s$^{-1}$ cm$^{-2}$) as responsive. The majority of units that passed this criterion showed increases in firing at higher irradiance, although a subset clearly

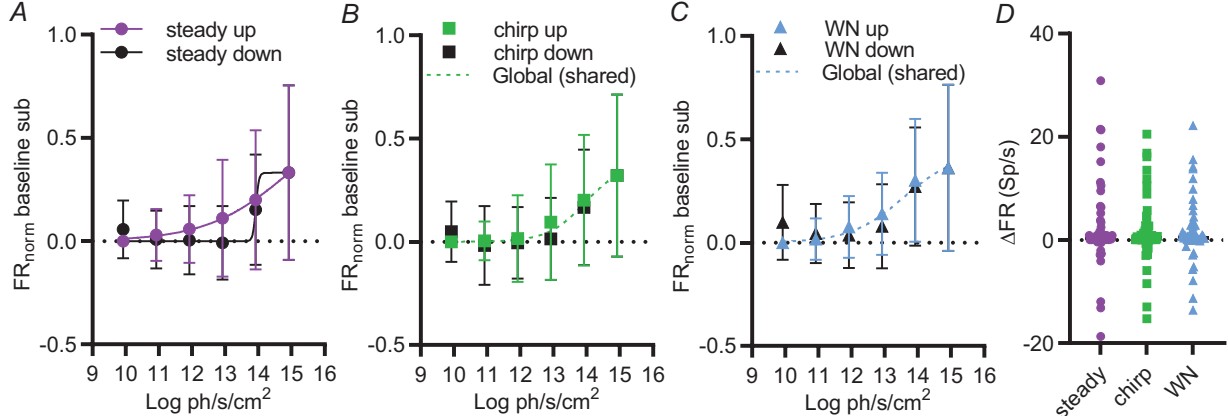

**Figure 3. Monotonic irradiance response relationships for the population of SCN light responsive units**
*A* to *C*, baseline subtracted, normalised firing rate, averaged across repeats of the staircase for individual units ($n = 57$), and across all units (mean $\pm$ SD) as a function of irradiance for ascending (up) and descending (down) phases (coloured *vs.* black points respectively) for steady (*A*), chirp (*B*) and WN (*C*) phases of the step. Curves show sigmoidal function fit to combined ascending and descending phases for chirp and WN, but separately for steady light (*F* test, steady: $P = 0.0256$, chirp: $P = 0.3097$, WN: $P = 0.3804$). Plots showing individual data points are included as Supporting information (Fig. S1). *D*, mean $\Delta$ firing rate between the highest and the lowest light levels under steady light, chirp and WN conditions for light responsive units (Friedman test, $P = 0.1421$, $F = 3.903$, $n = 57$). [Colour figure can be viewed at wileyonlinelibrary.com]

showed progressive reductions in firing consistent with the view that some SCN neurons are suppressed by light (Fig. 4).

We next attempted to classify single units according to how well their firing rate could be described by a monotonic relationship with irradiance. To this end, we applied a sigmoidal fit to a plot of firing rate *vs.* log(irradiance) for each single unit for each element of the irradiance ramp (up *vs.* down and steady, WN or

chirp). Because we were interested in identifying units with potentially unimodal tuning (progressive increases in FR across dimmer irradiances returning to baseline at higher irradiance), we screened for units with a relatively poor goodness of fit for this model. In practice, only four of the records with a reasonable modulation in firing rate were clear outliers in terms of goodness of fit ($r^2 <$ 0.85) (Fig. 5*A*, units circled). In each case, the poor fit was restricted to the down phase of the staircase stimulus

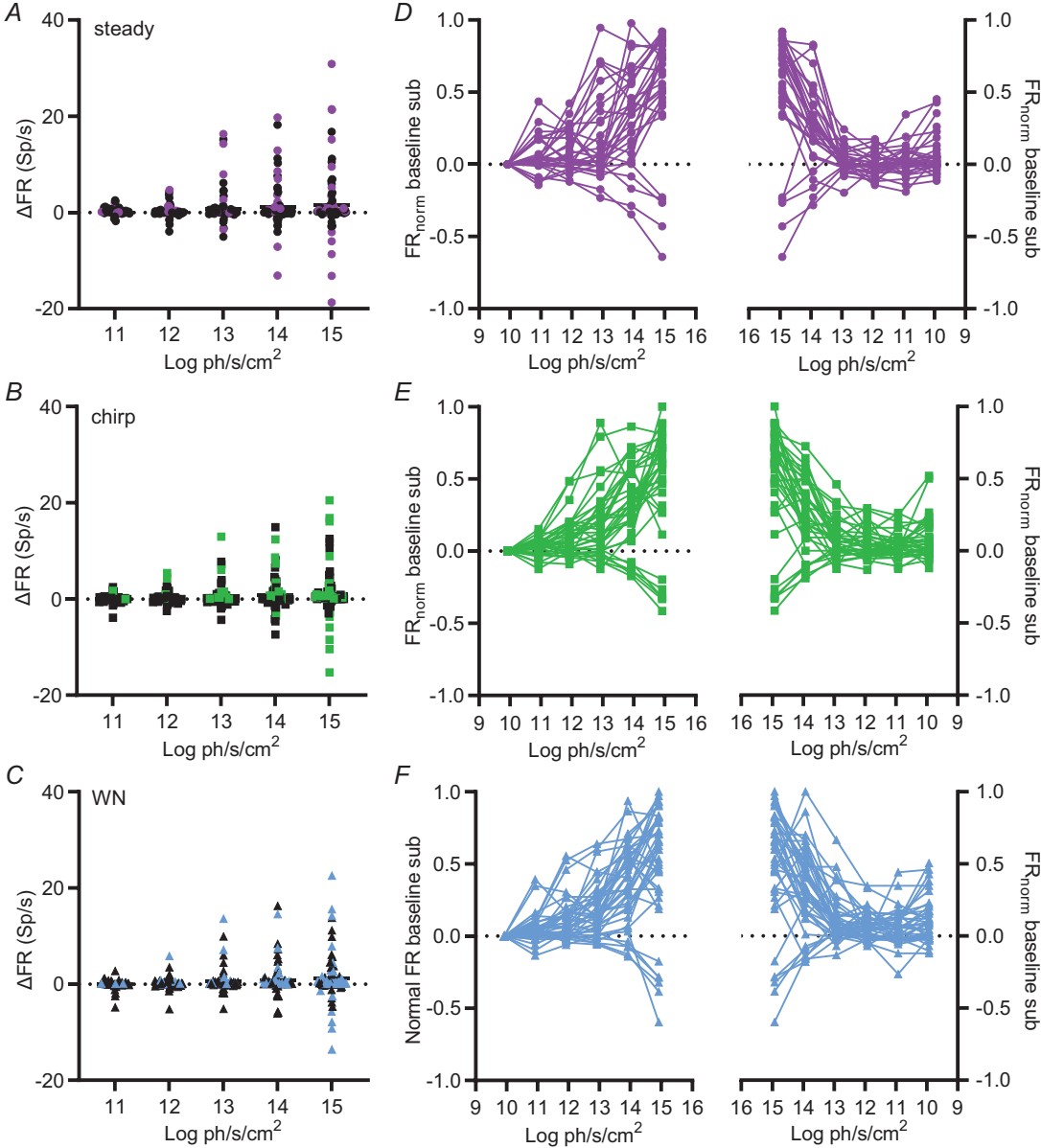

**Figure 4. Irradiance response relationships at single unit resolution**

*A* to *C*, distribution of mean ΔFR (change in FR at each step *vs.* that at dimmest irradiance) for all SCN units ($n = 84$) as a function of irradiance across ascending steps during (*A*) steady light, (*B*) chirp and (*C*) WN. Units in which ΔFR exceeded 2 SD of FR at the bottom step were classified as responsive to the irradiance staircase (coloured). *D* to *F*, mean normalised baseline subtracted FR as a function of irradiance for rising (left) and falling (right) phases of the staircase for all single units identified as responsive to the irradiance staircase, as in (*A*) to (*C*), under (*D*) steady ($n = 31$), (*E*) chirp ($n = 35$) and (*F*) WN ($n = 41$) phases of the step. [Colour figure can be viewed at wileyonlinelibrary.com]

and the biggest change in firing *vs.* baseline was achieved at the highest irradiance (Fig. 5*B*). On this basis, we conclude that they are not clear examples of unimodal irradiance coding. The remaining group that could in principle contain unimodal response types comprises those neurons suppressed by irradiance. Thus, if their firing rate was already excited at the dimmest element of the ramp, then a unimodal relationship could appear as progressive inhibition at higher irradiance. Plots of irradiance response relative to activity in the dark revealed

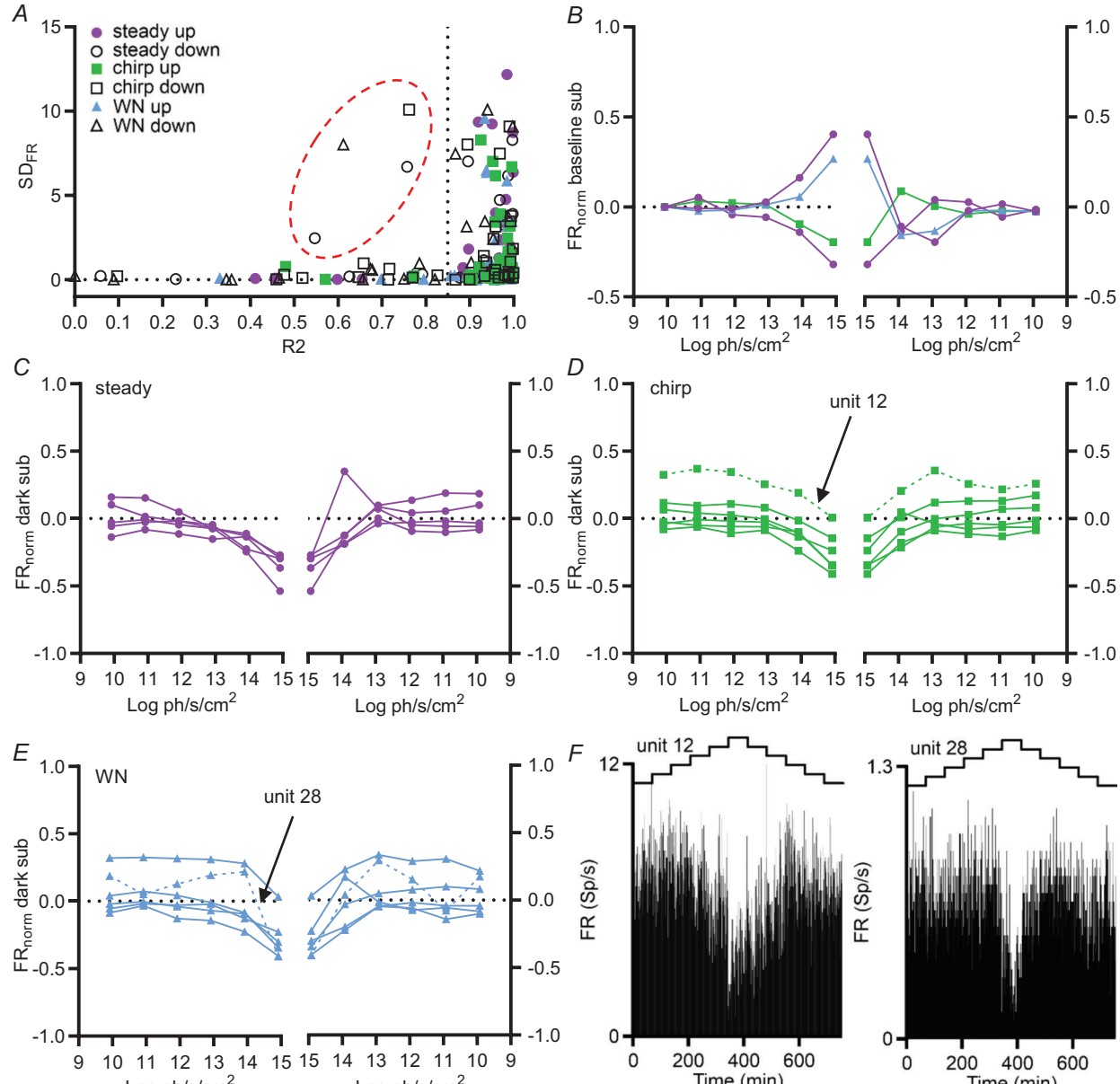

**Figure 5. Limited evidence for unimodal irradiance response relationships in the mouse SCN *in vivo***
*A*, distribution of standard deviation in FR across the staircase (SD$_{FR}$) *vs.* goodness of fit ($r^2$) for sigmoidal function for all units responsive to staircase stimuli for both ascending and descending staircase. A dotted vertical line shows threshold for defining activity as well fit by the sigmoidal function ($r^2 \geq 0.85$). A dashed ellipse captures records considered to represent potential unimodal tuning on the basis of having both relatively high SD$_{FR}$ and poor $r^2$. *B*, mean normalised baseline subtracted FR as a function of irradiance under the relevant condition for units encompassed within the ellipse in (*A*) under increasing (left) and decreasing (right) phases of the staircase. *C* to *E*, mean normalised dark subtracted FR as a function of irradiance under (*C*) steady light, (*D*) chirp and (*E*) WN conditions under increasing (left) and decreasing (right) phases of the staircase for units suppressed by irradiance. *F*, PSTHs (bin: 1 s) across the staircase irradiance stimulus (shown above) for two units (12 and 28) identified as potentially unimodal on the basis of having raised FR at dimmest irradiance *vs.* darkness. [Colour figure can be viewed at wileyonlinelibrary.com]

two units (Units 12 and 28) for which activity could match this profile (Fig. 5*C–F*). Neither had a clear unimodal profile, leaving the more parsimonious explanation that these are monotonic suppressed by irradiance cells for which higher firing at the dimmest irradiance compared to dark is explained by baseline drift.

Having failed to identify clear examples of unimodal tuning in the SCN, we turned to further exploration of monotonic irradiance response profiles. Anecdotally, we noted that irradiance response functions could appear quite divergent between single units even from the same recording and under the same stimulus conditions (Fig. 6*A*). For a more systematic description of this phenomenon, we extracted $EC_{50}$ and Hill slope for the firing rate *vs.* log(irradiance) fit of all data well fit ($r^2 \geq 0.85$) by the sigmoidal function ($n = 22/36$; 23/35 and 25/41 single units for steady, WN and chirp conditions, respectively). Among that population, there was substantial variation in sensitivity ($EC_{50}$) and slope under all stimulus conditions (Fig. 6*B–D*). These properties were, to some extent, stationary because there were significant correlations in both $EC_{50}$ and Hill slope for up and down phases of the ramp ($EC_{50}$: steady $r_s = 0.2959$, $P = 0.1422$; chirp $r_s = 0.5802$, $P = 0.0037$; WN $r_s = 0.3875$, $P = 0.0278$; Hill slope: steady $r_s = 0.4509$, $P = 0.0280$; chirp $r = 0.3900$, $P = 0.0658$; WN $r_s = 0.3069$, $P = 0.1356$). Within the irradiance range tested here, we see many units for which the sensitivity is so low that the curve fit applies an $EC_{50}$ close to the highest irradiance tested. Conversely, there were also many units with $EC_{50}$ well within the range tested, with the most sensitive responses having $EC_{50}$ three orders of magnitude lower ($\sim$11.5) than 14.92 log photons $s^{-1}$ cm$^{-2}$.

## Discussion

The primary objective of the present study was to determine the nature of irradiance coding in the SCN at the population level. We specifically explored the possibility that, when presented with continuous changes in irradiance, the SCN adopts a sparse irradiance code in which individual neurons with unimodal response profiles track limited portions of the irradiance range. Such behaviour has recently been reported for the M1 ipRGCs innervating the SCN (Liu et al., 2023; Milner & Do, 2017). Indeed, we find that mouse SCN irradiance response profiles are overwhelmingly monotonic. That is true for the massed (populational) activity of these nuclei, as well as for isolated single units.

A comparison between our data and that reported for M1 ipRGCs presented with a similar stimulus (Liu et al., 2023; Milner & Do, 2017) reveals both similarities and differences. The most important difference is that SCN neurons do not relax towards baseline firing rate at higher irradiances to match the silencing of many M1 ipRGCs under bright light. This suggests that the irradiance response of SCN neurons is not simply inherited from that of its retinal inputs. In other respects, however, the datasets are qualitatively similar. A feature of the heterogeneity in irradiance tuning of M1 cells is variation in sensitivity and dynamic range and we see the same in the SCN. For the steady part of the staircase (absent concurrent chirps or WN) $EC_{50}$ values for irradiance:response relationships differ by more than two orders of magnitude across SCN units, which is approximately equivalent to that reported for M1 ipRGCs under equivalent stimulus conditions (Milner & Do, 2017). Similarly, differences in slope for curve fits reveal that some units show graded changes in

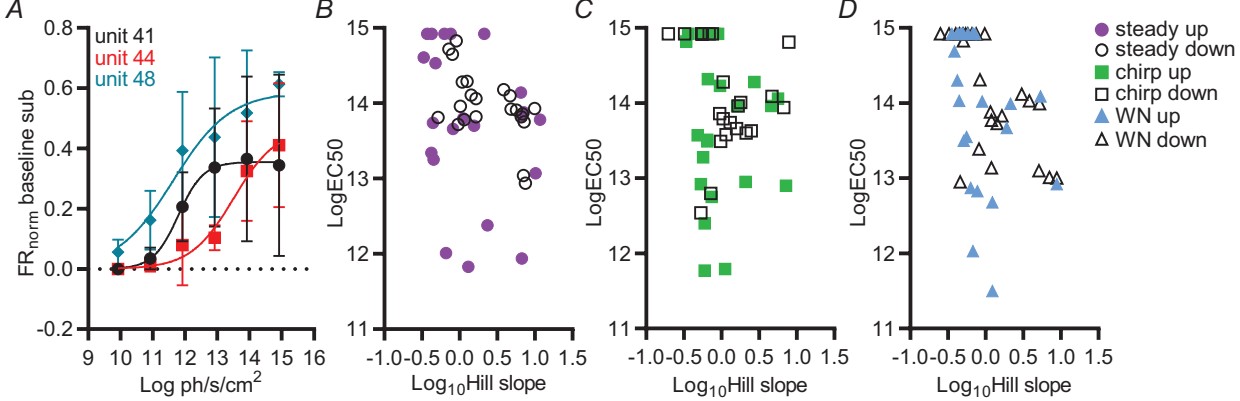

**Figure 6. Diversity in irradiance response relationships among SCN neurons**
*A*, irradiance response functions for simultaneously recorded single units (mean ± SD) under chirp stimulus. Distribution of log $EC_{50}$ *vs.* log$_{10}$ Hill slope for all cells classified as having a positive relationship with irradiance for ascending (coloured markers) and descending (black markers) staircase stimuli for (*B*) steady light, (*C*) chirp and (*D*) WN. [Colour figure can be viewed at wileyonlinelibrary.com]

firing over a wide irradiance range, whereas others have narrower tuning (again mirroring similar diversity in M1 population).

Variability in irradiance response relationships in the mouse SCN has implications for the nature of irradiance coding in this part of the brain. At a population level, SCN firing rates show progressive increases in firing across at least 1000× change in irradiance, although this masks substantial diversity at single unit level. The variation in slope and $EC_{50}$ in our dataset reveals differences in irradiance tuning across the population of light responsive units. In this way, the SCN does not have a perfectly distributed irradiance code in which the full stimulus would be represented by all neurons. Rather, our data indicate that it has elements of a sparse code, with individual units responding to different portions of the irradiance range. Such a sparse code could have the advantage of allowing some neurons to track the whole irradiance range (albeit with poor resolution), whereas others track a narrower range at higher resolution.

Our staircase stimulus spanned an irradiance range corresponding to that from dim moonlight to overcast day (Fig. 2*A-B*). This does not cover the full range of natural irradiances (nor that over which mammalian vision functions), but does cover the most informative irradiances for telling time of day. At the dimmest irradiance tested, SCN firing patterns were similar to that in darkness, indicating that it is unlikely that our dataset would have been substantially changed had we extended our stimulus range to dimmer light. Extending to higher irradiances could have been more informative. Irradiances up to 50× brighter would fall within that of natural daylight and many SCN units did not show clear saturation in firing rate at the highest irradiance we tested. It therefore remains possible that we would have observed progressive increases in firing at population level, and perhaps the appearance of SCN neurons tuned to very bright light, with an extended stimulus range.

One interesting aspect of the current dataset is the difference observed in the ability of the SCN to track changes in irradiance for steady light, chirp and WN. Sigmoidal irradiance curves for the steady light condition were not symmetrical for ascending and descending steps and this was observed at both population and single cell levels. This evidence for hysteresis mirrors some previous recordings for ipRGCs (Schmidt et al., 2014), including for the comparable staircase stimulus (Milner & Do, 2017). It could be argued that, if the main role of the SCN is to encode ambient light levels, then it should be able to do so irrespective of the direction of change. However, extended exposure to invariant light either presented as full-field steps or as part of a continuous staircase pyramid, is an unnatural stimulus. In active view, the light falling on receptive fields of individual visual neurons in the retina and brain is constantly changing. This originates not only as a result of modulations in ambient light, but also because of changes in the subject's direction of view across scenes containing spatial patterns. Accordingly, it is interesting that responses to up and down phases of the staircase pyramid were symmetrical in the presence of higher frequency modulations (WN and chirps). This suggests more reliable irradiance coding by the SCN in the presence of ongoing visual stimuli and fits, with a body of literature encouraging the use of more naturalistic stimuli in vision research (Karamanlis et al., 2022). We therefore propose that irradiance coding is best studied in the presence of superimposed fast modulations in light intensity.

The evidence for monotonic irradiance response functions in the SCN matches that previously reported for irradiance and staircase stimuli in the dorsolateral geniculate nucleus (Davis et al., 2015; Storchi et al., 2015), consistent with the view that it is a common form of irradiance coding in the brain. Nevertheless, the diversity of slope and $EC_{50}$ values across the SCN population reveals complexity at the single neuron level. The summed activity of SCN neurons achieves a simple monotonic increase in firing across a range of irradiances encompassing the twilight to daytime transition. However, most individual irradiance responsive SCN neurons code irradiance only over a restricted portion of this range. Future work could focus on explaining how this variability at single cell level relates to irradiance-dependent responses mediated by the SCN including circadian entrainment.

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

## Additional information

### Data availability statement

The data that support the findings of this study are available from the corresponding authors upon reasonable request.

### Competing interests

The authors declare that they have no competing interests.

### Author contributions

P.O.-F., R.J.L. and B.B.-O. were responsibe for the study concept and design. P.O.-F., R.J.L., B.B.-O. and F.P.M. were responsibe for data acquisition, analysis and interpretation. P.O.-F., R.J.L., J.R. and R.S. were responsibe for the statistical analysis. P.O.-F. and R.J.L. were responsibe for drafting the manuscript. All authors were responsibe for critical revision of the manuscript for important intellectual content. All authors have read and approved the final version of this manuscript and agree to be accountable for all aspects of the work in ensuring that questions related to the accuracy or integrity of any part of the work are appropriately investigated and resolved. All persons designated as authors qualify for authorship, and all those who qualify for authorship are listed.

### Funding

This study was funded by European Union's Horizon 2020 research and innovation programme under the Marie Sklodowska-Curie grant (Grant Code: 897 951) to PO-F; a Wellcome Trust Investigator award (grant code: 210 684/Z/18/Z) to RJL; and a Sir Henry Dale fellowship from Wellcome Trust (grant code: 220 163/Z/20/Z) to RS.

## Acknowledgements

We thank Professor Tim Brown (Manchester University) for his comments on the manuscript and Dr Annette Allen (Manchester University) for her help with setting up light stimuli.

## Keywords

electrophysiology, irradiance coding, mouse, staircase stimulous, suprachiasmatic nucleus

## Supporting information

Additional supporting information can be found online in the Supporting Information section at the end of the HTML view of the article. Supporting information files available:

**Statistical Summary Document**
**Peer Review History**
**Supporting Information**

