## [Peer Review History · The Journal of Physiology]

The mouse suprachiasmatic nucleus encodes irradiance via a diverse population of neurons monotonically tuned to different ranges of intensity.

Patrycja Orłowska-Feuer, Beatriz Bano-Otalora, Jessica Rodgers, Franck P Martial, Riccardo Storchi, and Robert J Lucas
DOI: 10.1113/JP285000

Corresponding author(s): Patrycja Orłowska-Feuer (patrycjaanna.orłowska-feuer@manchester.ac.uk)

The following individual(s) involved in review of this submission have agreed to reveal their identity: Hugh David Piggins (Referee #2)

Review Timeline:

Submission Date:	18-May-2023
Editorial Decision:	17-Jul-2023
Revision Received:	04-Aug-2023
Editorial Decision:	17-Aug-2023
Revision Received:	05-Sep-2023
Accepted:	11-Sep-2023

Senior Editor: Katalin Toth

Reviewing Editor: Nathan Schoppa

Transaction Report:

Dear Dr Orlowska-Feuer,

Re: JP-RP-2023-285000 "The mouse suprachiasmatic nucleus encodes irradiance via a diverse population of neurons monotonically tuned to different ranges of intensity." by Patrycja Orlowska-Feuer, Beatriz Bano-Otalora, Jessica Rodgers, Franck P Martial, Riccardo Storchi, and Robert J Lucas

Thank you for submitting your manuscript to The Journal of Physiology. It has been assessed by a Reviewing Editor and by 1 expert referee and we are pleased to tell you that it is acceptable for publication following satisfactory revision.

REVISION CHECKLIST:

We look forward to receiving your revised submission.

Yours sincerely,

Katalin Toth
Senior Editor
The Journal of Physiology

REQUIRED ITEMS

-Author photo and profile. First (or joint first) authors are asked to provide a short biography (no more than 100 words for one author or 150 words in total for joint first authors) and a portrait photograph. These should be uploaded and clearly labelled with the revised version of the manuscript. See Information for Authors for further details.

-The Journal of Physiology funds authors of provisionally accepted papers to use the premium BioRender site to create high resolution schematic figures. Follow this link and enter your details and the manuscript number to create and download figures. Upload these as the figure files for your revised submission. If you choose not to take up this offer we require figures to be of similar quality and resolution. If you are opting out of this service to authors, state this in the Comments section on the Detailed Information page of the submission form. The link provided should only be used for the purposes of this submission. Authors will be charged for figures created on this premium BioRender account if they are not related to this manuscript submission.

-Please upload separate high-quality figure files via the submission form.

-Please ensure that the Article File you upload is a Word file.

-A Statistical Summary Document, summarising the statistics presented in the manuscript, is required upon revision. It must be on the Journal's template, which can be downloaded from the link in the Statistical Summary Document section here: https://jp.msubmit.net/cgi-bin/main.plex?form_type=display_requirements#statistics

-Please include an Abstract Figure file, as well as the figure legend text within the main article file. The Abstract Figure is a piece of artwork designed to give readers an immediate understanding of the research and should summarise the main conclusions. If possible, the image should be easily 'readable' from left to right or top to bottom. It should show the physiological relevance of the manuscript so readers can assess the importance and content of its findings. Abstract Figures should not merely recapitulate other figures in the manuscript. Please try to keep the diagram as simple as possible and without superfluous information that may distract from the main conclusion(s). Abstract Figures must be provided by authors no later than the revised manuscript stage and should be uploaded as a separate file during online submission labelled as File Type 'Abstract Figure'. Please ensure that you include the figure legend in the main article file. All Abstract Figures should be created using BioRender. Authors should use The Journal's premium BioRender account to export high-resolution images. Details on how to use and access the premium account are included as part of this email.

-Please include a full title page as part of your article (Word) file (containing title, authors, affiliations, corresponding author name and contact details, keywords, and running title).

EDITOR COMMENTS

Reviewing Editor:

The suprachiasmatic nucleus (SCN) plays an important role in regulating a variety of physiological responses to environmental changes in light intensity. One unresolved question is how the vast 10^9 -fold change in irradiance across a day is encoded in the SCN. This study uses electrophysiological recordings in SCN in anesthetized mice to distinguish between essentially two hypotheses: a 'distributed code' in which a uniform population of neurons in SCN each expresses gradual changes in firing across the full irradiance range versus a sparse code in which individual neurons encode restricted irradiance ranges but differ in the irradiance range in which they are responsive. A key technical feature of the study is the use sophisticated visual stimuli (staircases in illumination given in the presence of white noise and chirps) that can approximate natural scenes. Such stimuli have been used previously to show that intrinsically photosensitive retinal ganglion cells (ipRGCs) that project to SCN have a sparse code. The study's main finding is that, when staircase stimuli are used, most single SCN neurons display monotonic changes in firing within a narrow range, but with differing ranges, consistent with the sparse code. Also, the neuronal response to irradiance is more reliable with the background of higher temporal frequency modulation in intensity than with steady light. The study has been reviewed by one expert reviewer and the reviewing editor. They felt that the study is addressing an important unresolved question with an innovative approach. The study was also considered to be well-designed and executed and well-written and clearly presented.

There were only a handful of minor concerns that were raised by the reviewer that will need to be addressed with changes in the text by the authors. In addition, I would raise the following additional minor concerns that should be addressed:

(1) The authors should, within the Methods section, summarize the statistical methods that were used to draw the conclusions that were made in the study (e.g., the F-test and Friedman test). The manuscript includes a Data Analysis section, but the section now only appears to describe the statistical methods that were used to isolate single units.

(2) The authors will need to be sure to include some of their detailed results in experiments in which $n > 30$ as Supporting Information. For example, Fig. 3A-C shows the mean{plus minus}SD without data points representing individual experiments, but plots showing the individual experiments should be included as Supporting Information.

(3) I would like to see a brief discussion in the Discussion section around the issue of the variable light intensity ranges in which the different SCN single units operated. The study begins by raising the question of how a 10^9 -fold change in irradiance is encoded in SCN, but the single units plotted in Fig. 6B-D show only a $\sim 10^3$ -fold difference in the EC50 values. The apparent mismatch does not appear to be unique to the present study - Milner and Do (2017) found similar moderate differences in the activation ranges in their recordings of single ipRGCs - but it does still leave open the question of how exactly the entirety of the 10^9 -fold change in irradiance is encoded. Do the authors, for example, think that there are neurons that are more rare (and not picked up in their recordings) that encode the more extreme ends of the irradiance range?

REFEREE COMMENTS

Referee #2:

How environmental light influences the activity of neurons of the brain's main circadian pacemaker, the suprachiasmatic nuclei (SCN), has and continues to be highly topical. There has been considerable advances in characterising the different types of retinal ganglion cells and in understanding how their outputs can regulate brain activity. However, the input to the SCN is non-imaging forming and it has remained unclear as to how the SCN neurons code for changing levels of irradiance. A factor in this is that many investigators have used unnatural visual stimuli and consequently this has restricted progress. In this nicely written and succinct manuscript, Orłowska-Feuer and colleagues use sophisticated visual stimuli (staircases in illumination given in the presence of white noise and chirps) that better approximate natural scenes and probe how SCN neurons at single cell and population level code for changes in irradiance. Using this original and innovative approach, the authors show that the monotonic relationship between irradiance levels of SCN neuronal activity is sustained with their staircase stimuli and that the neuronal response to irradiance is more reliable with the background of higher temporal frequency modulation in intensity than with steady light. When examining the ensemble population of SCN neurons, a wide range of irradiances, while some single units are responsive over a narrower and select range. This gives evidence to the

idea that sparse coding at single neuron level is used to build a much wider population level coding. This study is very well-designed and executed. The data set is impressive and the results/conclusions robust. This will be important to the circadian field and to the wider visual neuroscience community as it provides support for the drive to use more naturalistic stimuli to probe such questions.

END OF COMMENTS

Confidential Review

18-May-2023

Division of Neuroscience
AV Hill Building
School of Biological Sciences
The University of Manchester
Oxford Road
Manchester M13 9PL

Tel: +44 161 2755251
www.manchester.ac.uk

Professor Katalin Toth
Senior Editor
The Journal of Physiology

3 Aug 2023

Dear Professor Katalin Toth,

Thank you for provisional acceptance of our manuscript entitled "*The mouse suprachiasmatic nucleus encodes irradiance via a diverse population of neurons monotonically tuned to different ranges of intensity*" for publication in *The Journal of Physiology*. We were delighted that the reviewers found the paper to be of merit and we are grateful for the opportunity to publish our work in the *Journal*.

As requested, we have revised manuscript according to comments and marked changes across the manuscript (red font) and summarised implemented changes for each comment below (red font). This has resulted in adding information to Data analysis section, Discussion and submission of Supporting Information file showing individual data points for Fig. 3A-C.

We have also revised the manuscript text in terms of wording rather than context (all changes marked throughout the manuscript), have added a graphical abstract and the first author's biosketch and photo.

Yours sincerely,

Patrycja Orłowska-Feuer

Rob Lucas

EDITOR COMMENTS

Reviewing Editor:

The suprachiasmatic nucleus (SCN) plays an important role in regulating a variety of physiological responses to environmental changes in light intensity. One unresolved question is how the vast 10^9 -fold change in irradiance across a day is encoded in the SCN. This study uses electrophysiological recordings in SCN in anesthetized mice to distinguish between essentially two hypotheses: a 'distributed code' in which a uniform population of neurons in SCN each expresses gradual changes in firing across the full irradiance range versus a sparse code in which individual neurons encode restricted irradiance ranges but differ in the irradiance range in which they are responsive. A key technical feature of the study is the use sophisticated visual stimuli (staircases in illumination given in the presence of white noise and chirps) that can approximate natural scenes. Such stimuli have been used previously to show that intrinsically photosensitive retinal ganglion cells (ipRGCs) that project to SCN have a sparse code. The study's main finding is that, when staircase stimuli are used, most single SCN neurons display monotonic changes in firing within a narrow range, but with differing ranges, consistent with the sparse code. Also, the neuronal response to irradiance is more reliable with the background of higher temporal frequency modulation in intensity than with steady light. The study has been reviewed by one expert reviewer and the reviewing editor. They felt that the study is addressing an important unresolved question with an innovative approach. The study was also considered to be well-designed and executed and well-written and clearly presented.

Thank you.

There were only a handful of minor concerns that were raised by the reviewer that will need to be addressed with changes in the text by the authors. In addition, I would raise the following additional minor concerns that should be addressed:

(1) The authors should, within the Methods section, summarize the statistical methods that were used to draw the conclusions that were made in the study (e.g., the F-test and Friedman test). The manuscript includes a Data Analysis section, but the section now only appears to describe the statistical methods that were used to isolate single units.

The required information was added into the Method section.

(2) The authors will need to be sure to include some of their detailed results in experiments in which $n > 30$ as Supporting Information. For example, Fig. 3A-C shows the mean{plus minus}SD without data points representing individual experiments, but plots showing the individual experiments should be included as Supporting Information.

As required the version of Fig. 3A-C showing individual data points was included as Supporting Information.

(3) I would like to see a brief discussion in the Discussion section around the issue of the variable light intensity ranges in which the different SCN single units operated. The study begins by raising the question of how a 10^9 -fold change in irradiance is encoded in SCN, but the single units plotted in Fig. 6B-D show only a $\sim 10^3$ -fold difference in the EC50 values. The apparent mismatch does not appear to be unique to the present study - Milner and Do (2017) found similar moderate differences in the activation ranges in their recordings of single ipRGCs - but it does still leave open the question of how exactly the entirety of the 10^9 -fold change in irradiance is encoded. Do the authors, for example, think that there are neurons that are more rare (and not picked up in their recordings) that encode the more extreme ends of the irradiance range?

Thank you for pointing that out. A brief discussion was added as appropriate.

REFEREE COMMENTS

Referee #2:

How environmental light influences the activity of neurons of the brain's main circadian pacemaker, the suprachiasmatic nuclei (SCN), has and continues to be highly topical. There has been considerable advances in characterising the different types of retinal ganglion cells and in understanding how their outputs can regulate brain activity. However, the input to the SCN is non-imaging forming and it has remained unclear as to how the SCN neurons code for changing levels of irradiance. A factor in this is that many investigators have used unnatural visual stimuli and consequently this has restricted progress. In this nicely written and succinct manuscript, Orlowska-Feuer and colleagues use sophisticated visual stimuli (staircases in illumination given in the presence of white noise and chirps) that better approximate natural scenes and probe how SCN neurons at single cell and population level code for changes in irradiance. Using this original and innovative approach, the authors show that the monotonic relationship between irradiance levels of SCN neuronal activity is sustained with their staircase stimuli and that the neuronal response to irradiance is more reliable with the background of higher temporal frequency modulation in intensity than with steady light. When examining the ensemble population of SCN neurons, a wide range of irradiances, while some single units are responsive over a narrower and select range. This gives evidence to the idea that sparse coding at single neuron level is used to build a much wider population level coding. This study is very well-designed and executed. The data set is impressive and the results/conclusions robust. This will be important to the circadian field and to the wider visual neuroscience community as it provides support for the drive to use more naturalistic stimuli to probe such questions.

Thank you.

END OF COMMENTS

Dear Dr Orlowska-Feuer,

Re: JP-RP-2023-285000R1 "The mouse suprachiasmatic nucleus encodes irradiance via a diverse population of neurons monotonically tuned to different ranges of intensity." by Patrycja Orlowska-Feuer, Beatriz Bano-Otalora, Jessica Rodgers, Franck P Martial, Riccardo Storchi, and Robert j Lucas

Thank you for submitting your manuscript to The Journal of Physiology. It has been assessed by a Reviewing Editor and we are pleased to tell you that it is acceptable for publication following satisfactory revision.

REVISION CHECKLIST:

Please upload two versions of your manuscript text: one with all relevant changes highlighted and one clean version with no changes tracked. The manuscript file should include all tables and figure legends, but each figure/graph should be uploaded as separate, high-resolution files. The journal is now integrated with Wiley's Image Checking service. For further details, see: <https://www.wiley.com/en-us/network/publishing/research-publishing/trending-stories/upholding-image-integrity-wileys-image-screening-service>.

We look forward to receiving your revised submission.

Yours sincerely,

Katalin Toth
Senior Editor
The Journal of Physiology

EDITOR COMMENTS

Reviewing Editor:

The authors have adequately addressed the prior concerns that were raised by the reviewing editor.

Senior Editor:

Please, provide a citation for Kilosort (which version is used here?).

END OF COMMENTS

1st Confidential Review

04-Aug-2023

Division of Neuroscience
AV Hill Building
School of Biological Sciences
The University of Manchester
Oxford Road
Manchester M13 9PL

Tel: +44 161 2755251
www.manchester.ac.uk

Professor Katalin Toth
Senior Editor
The Journal of Physiology

5 Sep 2023

Dear Professor Katalin Toth,

Thank you for provisional acceptance of our manuscript entitled "*The mouse suprachiasmatic nucleus encodes irradiance via a diverse population of neurons monotonically tuned to different ranges of intensity*" for publication in *The Journal of Physiology*. We were delighted that the reviewers found the paper to be of merit and we are grateful for the opportunity to publish our work in the *Journal*.

As requested, we have added citation for Kilosort and specified version which we used.

Yours sincerely,

Patrycja Orłowska-Feuer

Rob Lucas

EDITOR COMMENTS

Reviewing Editor:

The authors have adequately addressed the prior concerns that were raised by the reviewing editor.

Thank you.

Senior Editor:

Please, provide a citation for Kilosort (which version is used here?).

We used Kilosort2 based on the paper *Kilosort: realtime spike-sorting for extracellular electrophysiology with hundreds of channels*. (<https://doi.org/10.1101/06148>) by Pachitariu M, Steinmetz N, Kadir S, Carandini M & Harris KD. (2016). This info has been added to the manuscript.

Dear Dr Orlowska-Feuer,

Re: JP-RP-2023-285000R2 "The mouse suprachiasmatic nucleus encodes irradiance via a diverse population of neurons monotonically tuned to different ranges of intensity." by Patrycja Orlowska-Feuer, Beatriz Bano-Otalora, Jessica Rodgers, Franck P Martial, Riccardo Storchi, and Robert j Lucas

We are pleased to tell you that your paper has been accepted for publication in The Journal of Physiology.

Authors should note that it is too late at this point to offer corrections prior to proofing. The accepted version will be published online, ahead of the copy edited and typeset version being made available. Major corrections at proof stage, such as changes to figures, will be referred to the Editors for approval before they can be incorporated. Only minor changes, such as to style and consistency, should be made at proof stage. Changes that need to be made after proof stage will usually require a formal correction notice.

Yours sincerely,

Katalin Toth
Senior Editor
The Journal of Physiology

P.S. - You can help your research get the attention it deserves! Check out Wiley's free Promotion Guide for best-practice recommendations for promoting your work at www.wileyauthors.com/eeo/guide. You can learn more about Wiley Editing Services which offers professional video, design, and writing services to create shareable video abstracts, infographics, conference posters, lay summaries, and research news stories for your research at www.wileyauthors.com/eeo/promotion.

IMPORTANT NOTICE ABOUT OPEN ACCESS: To assist authors whose funding agencies mandate public access to published research findings sooner than 12 months after publication, The Journal of Physiology allows authors to pay an Open Access (OA) fee to have their papers made freely available immediately on publication.

You can check if your funder or institution has a Wiley Open Access Account here: <https://authorservices.wiley.com/author-resources/Journal-Authors/licensing-and-open-access/open-access/author-compliance-tool.html>.